# New Frontiers in Three-Dimensional Culture Platforms to Improve Diabetes Research

**DOI:** 10.3390/pharmaceutics15030725

**Published:** 2023-02-22

**Authors:** Sundhar Mohandas, Vijaya Gayatri, Kriya Kumaran, Vipin Gopinath, Ramasamy Paulmurugan, Kunka Mohanram Ramkumar

**Affiliations:** 1Department of Biotechnology, School of Bioengineering, SRM Institute of Science and Technology, Kattankulathur 603203, Tamil Nadu, India; 2Department of Radiology, Molecular Imaging Program at Stanford, Canary Centre for Cancer Early Detection, Bio-X Program, Stanford University School of Medicine, Palo Alto, CA 94304, USA; 3Molecular Oncology Division, Malabar Cancer Centre, Moozhikkara P.O, Thalassery 670103, Kerala, India

**Keywords:** Diabetes mellitus, three-dimensional culture system, disease model, β-cell cultivation

## Abstract

Diabetes mellitus is associated with defects in islet β-cell functioning and consequent hyperglycemia resulting in multi-organ damage. Physiologically relevant models that mimic human diabetic progression are urgently needed to identify new drug targets. Three-dimensional (3D) cell-culture systems are gaining a considerable interest in diabetic disease modelling and are being utilized as platforms for diabetic drug discovery and pancreatic tissue engineering. Three-dimensional models offer a marked advantage in obtaining physiologically relevant information and improve drug selectivity over conventional 2D (two-dimensional) cultures and rodent models. Indeed, recent evidence persuasively supports the adoption of appropriate 3D cell technology in β-cell cultivation. This review article provides a considerably updated view of the benefits of employing 3D models in the experimental workflow compared to conventional animal and 2D models. We compile the latest innovations in this field and discuss the various strategies used to generate 3D culture models in diabetic research. We also critically review the advantages and the limitations of each 3D technology, with particular attention to the maintenance of β-cell morphology, functionality, and intercellular crosstalk. Furthermore, we emphasize the scope of improvement needed in the 3D culture systems employed in diabetes research and the promises they hold as excellent research platforms in managing diabetes.

## 1. Introduction

Diabetes mellitus (DM) is a chronic metabolic disorder primarily characterized by high blood glucose levels. DM has become a global health issue, with over 536 million people affected in 2021, representing ~6.8% of the world population. Currently, an estimated 6.7 million deaths occur due to diabetes every year. It is predicted that in 2045 around 783.2 million people will have diabetes. Currently, 39.4 million people remain undiagnosed, with the proportion of undiagnosed cases as high as 53.1%. Around the globe, ~USD966 billion are spent on health care expenditures yearly due to diabetes [1]. Diabetes is broadly classified into type 1 diabetes mellitus (T1DM), type 2 diabetes mellitus (T2DM), and gestational diabetes mellitus [2]. Around 90–95% [3] of people with diabetes are usually affected by T2DM and 5–10% with T1DM [4]. Both T1DM and T2DM are associated with micro- and macrovascular complications, including atherosclerosis, hypertension, neuropathy, retinopathy, and nephropathy. When left without attention, diabetes can lead to the deterioration of multiple organ systems and eventually lead to death. Physiologically relevant models that mimic human diabetic progression are urgently needed to identify new drug targets, drugs, and therapies.

Even though there has been remarkable progress in diabetic research and practice over the last decade, there is still a lack of reproducible and relevant models available to investigate pancreatic β-cell dysfunction in various diabetic conditions. Additionally, the progress in islet auto-transplantation has been hindered by the limited availability of primary human β-cells/islets. Furthermore, donor diversity and the variance in islet quality restrict the development of in vitro models for diabetes research. Traditional 2D cell-culture models, where monolayer cells are grown on a flat surface, have been extensively used to understand the pathological alterations in the background of high-glucose and high-lipid environments [5]. While the current 2D models are convenient to set up, they are limited in their capability to replicate the in vivo microenvironment where several layers of multiple cell types are involved in DM. In particular, due to the lack of cellular contact and extracellular matrix interaction, the monolayer 2D cultures cannot mimic the intracellular signalling, gene expression, and phenotypic fate that occur in vivo.

Moreover, it is impossible to mimic tissue-specific architecture, dynamic physiological and biochemical processes, and cell-to-cell communications using conventional 2D cultures. Deranged cellular organization and polarity, coupled with alterations in the mechanical properties of cells, including shape, stiffness [6,7], and smoothness, have been extensively reported in 2D cell cultures [8]. While animal models bridge some of the limitations observed in 2D cell cultures, they are accompanied by a few challenges. The use of animals in studies raises major ethical concerns. Moreover, the use of rodent models is associated with low predictive levels of human therapeutic responses. Further, the variance in the genomic makeup of animals has encouraged researchers to develop in vitro models that replicate the physiological microenvironment in human diabetes.

Three-dimensional (3D) cultures have been reported to be first utilized in the early 20th century. Over the last decade, this technology has grown considerably to incorporate new cell lines [9] and media, which has facilitated computational imaging and simulation capabilities [8]. The ability to perform experiments in an in vivo environment using an in vitro system has bridged the gap between the 2D models, animal models, and human clinical trials. The 3D cell-culture models have exhibited distinct advantages that allow the investigator to control the cellular environment without altering the properties of the cells in the micro-tissues while also accounting for cell–cell and cell–ECM interactions using multiple cell types. Three-dimensional cell-culture techniques offer incredible potential for applications to regenerative medicine and drug discovery in DM [5]. Further, the integration of biosensors into the 3D systems also provides the opportunity for non-invasive and real-time monitoring of cell viability, proliferation, migration, and drug susceptibility. In this review manuscript, we aim to extensively summarize the various research activities currently in progress in 3D cell-culture models and their applications in diabetic research.

## 2. Current Experimental Models of Diabetes

Experimental models of various types have been extensively employed in diabetes research to understand the disease mechanisms involved and to be utilized as platforms to identify (screen), characterize, and test the efficiency and toxicity of compounds exhibiting therapeutic potential. With regards to diabetes, both in vitro models employing cells, as well as in vivo models employing animals, have been widely demonstrated. Specific models have been described and used to replicate the pathogenesis based on the etiology of T1DM, T2DM, and the associated complications. In vitro models have been readily adapted for the high-throughput screening of a vast compound library, which were then tested in animal models to see if the therapeutic activity could replicated under in vivo conditions [10].

The in vitro models include the conventional 2D cell-culture models where cells are grown in a monolayer, whereas in 3D cell-culture models, an environmental constituent is introduced to simulate the contact between cells and ECM, which mimics in vivo conditions. Even though the conventional 2D models are simple and cost-efficient to establish, they are associated with glaring concerns, including alteration of gene expression and cell functionality throughout the experiment. Genetically engineered rodents have been utilized to exhibit spontaneous T1DM. Similarly, chemical, surgical, and genetic interventions on animals have been used to mimic the metabolic disorder, insulin resistance, and β-cell dysfunction to study T2DM [10]. Three-dimensional models of diabetes are promising to overcome the insufficiencies of conventional 2D systems while also providing in vivo relevance while overcoming the ethical concerns observed in animal studies.

### 2.1. 2D Models in Diabetic Research

Two--dimensional models using monolayer cell cultures are still being extensively used in the field of diabetic research due to their ease of incorporation in workflows, lesser time to obtain working cultures, wide availability of cell lines, simplicity in performance, and interpretation of functional tests and imaging analysis, as well as low-cost maintenance [11]. It should be noted that, in the 2D culture microenvironments, the polarised cells are exposed to the medium from one side only, and only a fragment of the cellular surface is in contact with the neighbouring cells compared to native tissues. Several studies have investigated the functionality of islet cells using 2D cultures using various extracellular matrix (ECM) components, including laminin, collagen, and fibronectin [12,13,14,15]. Nevertheless, modifications in gene expression, mRNA splicing, and the biochemistry of cells continue to limit the efficiency of these models. Moreover, other limitations, such as marked alterations in the cellular morphology, loss of polarity, and the method of proliferation, have been encountered while using traditional 2D cell cultures [11,16,17]. Rosenberg et al. identified that following islet isolation, the β-cells underwent marked negative alterations in structure and function when grown in 2D cultures. Notably, following isolation and cultivation in 2D culture, an absence of the peri-insular basement membrane was observed, along with the downregulation of integrin αvβ5 expression. Moreover, a significant increase in the apoptotic index and a marked elevation in DNA fragmentation were also identified on day 5 of β-cell culture in the 2D environment [18].

Furthermore, due to the deprivation of cell–cell and cell–environment interaction, the 2D cultures mimic neither the natural structure of the tissue nor the tissue microenvironment. There is also unlimited access to oxygen, nutrients, metabolites, and signalling molecules, which is in stark contrast to in vivo conditions with variable access to these components. Notably, the widely used 2D pancreatic culture models are unable to replicate the dynamic of insulin secretion in response to a stimulus of glucose. Further, they are posed with inherent challenges in maintaining the viability of β-cells over extended periods. Indeed, β-cells are susceptible to the cultivation method and have exhibited increased necrosis and loss of the ability to secrete insulin on flat 2D substrates. Researchers have documented that the lack of matrix–integrin and intercellular interactions led to increased apoptosis of β-cells followed by islet isolation [12,19,20].

### 2.2. Animal Models in Diabetic Research

Various animal models have enormously contributed to our understanding of the pathophysiology of both T1DM and T2DM (Table 1). The ease of manipulating these models through chemical, surgical, and genetic intervention presents certain advantages of having disease phenotypes with a moderate degree of similarity to humans. However, it is evident that due to species-specific differences, the fundamental genetic and molecular mechanisms significantly differ between humans and other animals. Indeed, despite similarities in disease phenotype, several structural and physiological differences have been documented in the islets of Langerhans between humans and rodents. Notably, a difference in islet architecture between humans and rodents has been documented, which raises concerns regarding the relevance and interpretation of data obtained from rodent studies to humans. These differences extend to marsupials and some nonhuman primates, which have been reported to exhibit inverted β-cell-to-other-endocrine-cell ratios [21,22,23,24]. Overall, these differences account for the functional coupling between β-cells of different species and explain the unique and complex dynamics of insulin secretion due to the inherent physiological disparity of humans with other species.

Animal models of diabetes involve either the spontaneous development of diabetes or the induction of diabetes through chemical, surgical, and genetic manipulation techniques. Spontaneous models include non-obese diabetic (NOD) mice, AKITA mice, BioBreeding (BB) rats, Long-Evans Tokushima Lean (LETL) rats, Komeda diabetes-prone (KDP) rats, and Lewis insulin-dependent diabetes mellitus (LEW-IDDM) rats, which have been among the most commonly utilized models for T1DM or autoimmune diabetes (Table 1). Since T1DM is caused by a lack of insulin due to β-cell destruction, this deficiency has also been induced in mouse models through chemical ablation of β-cells. Many subtypes and features of both T1DM and T2DM can be induced through the administration of diabetogenic agents such as streptozotocin, alloxan, and dithizone, which cause β-cell toxicity in the study animals. However, efficient administration of these toxins to animals is associated with a marked mortality rate. Moreover, whether hyperglycemia is developed due to the direct cytotoxic action on β-cells rather than as a consequence of insulin resistance should be considered. Many chemically induced models are also associated with damage to other organs, independent of diabetic pathology.

Both obese and non-obese models are used to study insulin resistance and β-cell failure in T2DM. While supplementation with a high-fat and/or high-sucrose diet has been employed to induce obese diabetic models, polygenic non-obese models using Goto-Kakizaki (GK) rats have been developed through multi-generational selective inbreeding of mildly glucose-intolerant Wistar rats. A major disadvantage with such congenic rat models is the requirement of extensive time and resources needed for breeding the animals for multiple generations [21,25,26]. Moreover, these models do not reproduce all the features of the complex diabetic phenotype observed in humans, thereby hindering the translational application of results collected using this model. Due to the differences in the physiology, metabolism, and genetics between rodent models and humans, the relevance of preclinical assessments in terms of both safety and efficiency of therapeutics is often controversial and limited.

**Table 1 pharmaceutics-15-00725-t001:** Experimental animal models for diabetes and its related complications.

**T1DM**			
**MODEL**	**ONSET**	**FEATURES**	**DISADVANTAGE**
NOD mouse	10 weeks	Closest representation to spontaneous autoimmunity of human T1DM. Similar genetic and environmental components to that of humans. Most widely used model for invasive, preclinical, and translational studies. Incidence is higher in females [27].	Various aspects of the disease cannot be studied due to the lack of understanding of T1DM in NOD mice and humans. Drugs effective in NOD mice were found to be ineffective in humans. Problems in translating dosage for drugs from NOD mice to humans. The NOD mice are susceptible to microbial exposure [28]. Occurrence of insulinitis. Gene–gene interactions and epistasis trigger some of the T1DM genes. The immune modulations found in NOD mice may be limited to only a few subtypes seen in human T1DM. Multiple experiments with a large number of models have to be performed to attain adequate capacity as the IDD loci in NOD models are strongly suppressed. The results of clinical trials using agents that could cure mouse diabetes were not satisfactory in humans [25].
BB rat	8–16 weeks	Widely used model to study the environmental effects of T1DM. Derived from two inbred types—DP-BB/Wor and DR-BB/Wor. Predominance of Th1-type lymphocytes. Similar to human T1DM in terms of characteristics. Develops with equal frequency in both males and females [27].	Profound T-cell lymphopenia is found to be problematic as it is not a property of T1DM. Animals exhibit lymphocytic thyroiditis and thymic epithelial cell defects. Levels of IAA, GAD-65A are lower than that exhibited in humans. IA2A is completely absent [27]. Inbreeding has led to polymorphisms and unpredictable autoimmune characteristics in the models [29]. Mechanism leading to β-cell auto-reactivity is not completely understood. No humoral reactivity with β-cells.
LETL rat	8–20 weeks	An inbred congenic strain which resembles the pathophysiology of T1DM in humans. No gender differences in the incidence and has two sub-strains. Characteristic elevation in plasma glucose levels and lymphocyte infiltration.	Most animals die within 30 weeks if proper insulin therapy is not given. Incidence of diabetes is shown only by 20% of animals. Patterns of serological or MLR reactivity are not affected by class II genes. The numbers of splenic lymphocytes are not significant.
LEW-IDDM rat	6–12 weeks	No gender differences in the incidence Arise spontaneously in a colony characterised by a defined MHC haplotype [30]. Swift progression of insulitis followed by β-cell destruction	Antibodies to GAD, IA-2 are present in humans but are not found in this model. Response to the environment and immunosuppression are unknown [30]. Autoimmunity is restricted to pancreatic β-cell [31].
KDP rat	8–16 weeks	No gender differences in the incidence Autoimmunity is regulated negatively [30].	Critical problem owing to the poor reproductive ability of diabetic animals Response to the environment and immunosuppression are unknown [30]. Not appropriate for preclinical prevention studies as there is no lymphopenia [32]. The pro-inflammatory cytokine profile is different when compared to humans.
AKITA mouse	3–4 weeks	Developed from a C57BL/6NSlc mouse through spontaneous mutation in the insulin 2 gene. Development of T1DM within 3–4 weeks. Used to investigate ER stress in β-cells and may be used to study diabetic nephropathy [33].	Untreated homozygotes rarely survive longer than 12 weeks. Gender-dependent, as T1DM is less prominent in females [34]. These models, due to their genetic makeup, could also develop kidney injuries to some extent as the mechanism of the mesangial matrix is not properly understood [28]. Abnormal cardiac function with nerve density loss has been identified [33].
**T2DM**			
**MODEL**	**ONSET**	**FEATURES**	**DISADVANTAGE**
Obese (ob/ob) Mouse	15 weeks	Monogenic. Mutations in leptin gene lead to obesity. Incidence of insulin resistance. Mild elevation in blood glucose levels [35].	Infertility in animals. High cost [36]. Toxic effects of fat that occur during obesity may lead to cardiac dysfunction [37]. Models of leptin deficiency are limited in the degree of fibrosis [38].
Diabetes (db/db) mouse	14 weeks	Monogenic. Modifications of C57BL strain. Incidence of morbid obesity. Incidence of acute insulin resistance.	Reproductive failure. Limited translatable insight. Uninephrectomy leads to diabetic nephropathy and display of severe tubular atrophy and tubulointerstitial fibrosis [36]. Reduced capacity of the db/db mouse to make a thermogenic response [39].
Zucker fa/fa rat	10 weeks	Monogenic and can exhibit β-cell degranulation and increased β-cell death. Inherited by males, which become diabetic between 8 and 10 weeks. Characterised by homozygous (fa/fa) mutation of the leptin hormone receptor. High lipid levels can be induced.	An important issue is infertility in male rats. Does not replicate the complete background pathogenic features of T2D. The pathway behind the failure of cell mass expansion is not fully understood. Islet pathology differs from humans [40,41].
KKAy (KK) mouse	8 weeks	Polygenic. Mildly obese and hyperleptinemic strain. Incidence of severe hyperinsulinemia. Prominent changes in pancreatic islets.	The mutation responsible for the KK phenotype is unknown [41]. Abnormalities of renal structure with severe decline in renal function [36]. Differ both genetically and phenotypically from each other in types [42].
New Zealand Obese (NZO) mouse	22 weeks and later	Polygenic Obesity-induced Pathogenetic mechanisms that lead to hyperglycemia are similar in mice and humans.	Level of insulin is lower than that in other obese models [42]. NZO islets lack the ability to initiate cell-cycle progression and to maintain the survival pathway [43]. More sensitive to high-fat diet-induced obesity [44].
Otsuka Long-Evans Tokushima Fat (OLETF) rat	18 weeks	Polygenic Exhibit mild obesity Incidence of late-onset hyperglycemia. The pancreatic islets undergo three stages of histological change. Diabetes is inherited by males.	Exhibit renal complications. Deficit in their ability to limit the size of meals [45].
Nagoya–Shibata–Yasuda (NSY) mouse	48 weeks	Polygenic and obesity-induced. Develops the disease in a sex-dependent manner, with higher incidence in females. Develops in an age-dependent manner.	Phenotypic characteristics and pathogenesis are mostly unknown. Diabetes in NSY mice is not caused by autoimmunity [46]. Obesity is not a major feature in these animals [47].

## 3. Three-Dimensional Cell-Culture Models

Three-dimensional cell-culture models create a more complex environment without the physical constraints faced in monolayer cultures, enabling cell populations to interact dynamically and supporting complicated cellular proliferation and maturation as observed in vivo [48]. Three-dimensional cell-culture models are a rapidly developing area of biotechnology that offers an incredible potential that could allow the implication of the 3R principle of reducing, refining, and replacing animal experiments. Scientists thus far have been able to fabricate working 3D models of skin, cornea, blood vessels, thyroid, liver, stomach organoids, bone, and cartilage. The establishment of pancreatic 3D models is gaining ground with potential applications in diabetic disease modelling and cell therapy in regenerative medicine.

The stability of immortalized β-cell cultures deteriorates over time, principally owing to phenotypic alterations generated by frequent passaging and unstable long-term culture. Therefore, in this context, 3D-based models have been considered alternatives to conventional 2D systems [49] and would enable uninterrupted long-term dynamic cell growth while mimicking in vivo tissue architecture. This eliminates the need for passaging while facilitating uninterrupted cell–cell interactions and developing more tissue-specific morphologies with a representative β-cell physiology [50]. Diabetes is a disease that affects several organs, and 3D models such as spheroids, organoids, bioreactors, and bioprints provide an opportunity to fabricate systems with specific features that more precisely replicate the pathophysiology of diabetic complications [51].

Transplantation of functional β-cells is a therapeutic approach currently hindered by the considerable difficulty in generating a sufficient number of β-cells ex vivo and, after that, in ensuring the viability of these cells at the site of transplantation. β-cells are susceptible to hypoxia and undergo rapid apoptosis or damage due to the host immune system. In this regard, human mesenchymal stem/stromal cells (hMSCs) have been used to improve the viability and function of β-cells in vivo and in vitro by establishing cell–cell contact and the secretion of trophic factors. However, the current co-cultivation concepts with β-cells cannot exploit the advantageous properties of hMSCs. β-cells grown in a 3D environment have been demonstrated to exhibit improved functionality and survival rates compared to 2D cultures. Various strategies have been reported to enable the large-scale cultivation of β-cells [52] (Table 2). These strategies have also been employed to create efficient diabetic disease models. Generally, a 3D cell-culture model can either be based on a scaffold or in a scaffold-free system. However, it should be noted that some of the 3D models have the flexibility of being incorporated in both scaffold and scaffold-free systems.

## 4. Scaffold-Based 3D Systems

Scaffold systems are more widely used due to the distinct advantages of having greater specificity and physiological relevance to the in vivo components in terms of varying porosity, surface chemistry, and permeability. Structural scaffolds usually comprise biopolymers that mimic the ECM in the physiological state. Scaffold systems can be synthetic or biological, with the most commonly used cross-linked polymer-based hydrogel and Matrigel scaffolds. Moreover, scaffolds based on nanofibers, collagen sponges, agarose-peptide microgels, polystyrene, and polycaprolactone are the other scaffold systems developed for various applications (Figure 1).

Interestingly, recent advancements in bioengineering have enabled the design of scaffolds as inserts that can be integrated into workflows of conventional cell-culture models. Various fabrication techniques, including 3D printing, particle leaching, and electrospinning, have been employed to obtain scaffolds. They are already being used extensively since they are predicted to facilitate a relatively close drug-discovery process [59]. Notably, scaffolding facilitates the interactions between cells and their extracellular matrices, encouraging the growth of regenerative cellular organizations [71]. Indeed, various studies have been performed using porous scaffolds such as PGA and PLGA for T1DM, highlighting the importance of cell–matrix interactions in mimicking the in vivo functionality of pancreatic islet cells [72].

A novel scaffold-based 3D cell-culture model has been employed by Xu et al. to recreate functional insulin-producing cells (IPCs) using stem cells [73]. Under 2D conditions, human dental pulp stem cells (hDPSCs) were first stimulated using small-molecule compounds including Activin A and Noggin to assist their differentiation into distinct endoderm-like cells and pancreatic-progenitor-like cells. These cells were then embedded in Matrigel to obtain IPCs that secrete insulin and c-peptide [73]. Another study by Liu et al. demonstrated the differentiation of human embryonic stem cells (hESCs) into mature pancreatic endocrine cells (PECs). Notably, in contrast with 2D cell culture, only 3D cultures facilitated the pancreatic specification efficiency and enhanced functional maturation. Additionally, they identified that 3D cell culture enhanced the commitment of stem cells through the obstruction of focal adhesion kinase (FAK)-dependent induction of the SMAD2/3 pathway and the upregulation of Connexin 36 (Cx36) expression [74]. More studies exploring the key signalling events behind the maturation and development of functional β-cells could further improve the reproducibility of 3D systems.

Luo et al. demonstrated a drug-screening model in diabetes utilizing a 3D culture of an insulinoma cell line (INS-1 cells) exposed to an elevated glucose concentration. Interestingly, the high glucose concentration initially promoted the INS-1 cell proliferation and insulin secretion, followed by the loss of both cell proliferation and secretion function. Indeed, a similar pattern of events was observed in hyperglycemic pathology, which provides additional validation to this model. They also developed a microfluidics-based circular drug concentration gradient generator integrated into the same micro-device along with the high-glucose 3D INS-1 cell model to test the utility of this device in screening anti-diabetic drugs [75]. Different scaffolding materials have demonstrated varying benefits, which could be applied to replicate specific disease outcomes. To investigate skin tissue regeneration, particularly in diabetic foot injury, Intini et al. fabricated porous 3D printed chitosan (CH)-based scaffolds [76]. A skin-like layer was obtained after co-seeding normal human dermal fibroblasts (NHDF) and keratinocytes (HaCaT) into the pores in the scaffolds. They employed 3D bioprinting to fabricate the porous microstructures, which was critical for the growth and interaction of fibroblasts and keratinocyte cells. Additionally, they used the 3D-printed scaffolds and demonstrated that these scaffolds improved wound healing in STZ-induced diabetic rats compared with the commercial patch and spontaneous healing [76]. Feng et al. developed and demonstrated a novel application of spheroids through the fabrication of an injectable non-cross-linked hyaluronic-acid gel containing spheroids [77]. This technique could show immense potential in producing clinically relevant therapeutic microspheroids with amplified sternness that could be injected directly into diabetic wounds.

Despite the distinct advantages that scaffold-based 3D systems offer, a range of factors needs to be optimized before their broad utilization in diabetic research [71]. Because the composition of the commercially available biological matrices is neither completely known nor clearly understood, they might contain unwanted contaminants and growth factors, which might disrupt the reproducibility and accuracy of the study model. Amongst natural scaffolds, currently Matrigel alone is showing great versatility [11]. Moreover, there seems to be variation across each scaffold when constructed, further highlighting the importance of duly choosing the appropriate 3D cell-culture model to increase its efficiency in diabetic research.

### 4.1. 3D Scaffold Systems with Integrated Biosensors

There is an inherent difficulty in achieving dynamic monitoring of cell behaviour through the conventional 2D imaging techniques, owing to the thick samples in the 3D cell-culture systems [78]. This problem warrants the development of novel sensing techniques that adopt biochemical, electrochemical, and optical sensors that can be integrated with the existing 3D systems to facilitate the measurement of real-time information on cell behaviour [79]. In this regard, Pan et al. developed a 3D electric cell/Matrigel substrate impedance sensing (3D-ECMIS) system to facilitate the real-time monitoring of cell viability and susceptibility to drugs in a non-invasive manner within the 3D scaffold [78]. This setup consisted of culturing Matrigel-encapsulated HepG2 cells in a 3D-ECMIS chip with gold electrodes and a portable multi-channel system to monitor the Matrigel construct. The staining of live/dead cells was performed to further validate the performance of the 3D-ECMIS system [78]. Such a 3D-based cell biosensing system has also been integrated with 3D hydrogels to obtain information on cell behaviour, which was previously unavailable through 2D-based imaging techniques [80]. Lee et al. developed and demonstrated a 3D capacitance biosensor equipped with vertically aligned electrodes to measure the change in capacitance and thereby obtain real-time data on chemosensitivity and viability of cells in the 3D system. The vertical alignment of the electrodes at different heights in the 3D capacitance sensor facilitated the measurement of cell proliferation, apoptosis, and cell migration using human breast cancer cells expressing GFP encapsulated in an alginate hydrogel system [80].

### 4.2. Organoid Models

Organoids are a three-dimensional, miniature form of an organ maintained in vitro. Organoids are considered layered systems that can improve the differentiation and maturation of epithelial cells by removing the apical layer from the air–liquid interface. Organoids can comprise several types of cells, which are obtained through culturing cells sourced from adult tissues, embryonic stem cells, or iPSCs [51]. Notably, due to the self-renewing and differentiating capabilities of cells, they self-organize into three-dimensional multilayer structures. Notably, the organoids are reported to organize at a similar frequency as observed in physiological conditions. There are two types of organoid structures: spheroids and extracellular matrix layered cultures. Some models have been made using a combination of these two. However, the distinction between organoids and spheroids is based on the cell source and attachment. While spheroidal cultures comprise free-floating cell aggregates obtained using ultra-low attachment plates, organoids are derived from stem cells embedded in an ECM hydrogel matrix (Figure 2). Owing to their complexity, organoids represent the in vivo environment better when compared to spheroids.

ECM and relevant growth factors facilitate the scaffold required for cell attachment and growth during the formation of the organoid layers (Figure 2). Organoids have been employed to mimic the function of an organ either in its physiological state or in the background of pathological cues such as hyperglycemia. There is an added advantage of cryopreserving the organoids as biobanks, which can be expanded indefinitely owing to the intrinsic capability of stem cells to renew and self-organize. An increased adaptation of adult stem cells in the workflow could avoid ethical issues such as that seen in using ESCs.

Pancreatic progenitor cells obtained from healthy or diseased tissues could be mixed with Matrigel or collagen matrix in order to mimic the functionality of the pancreas. The organoid models would further the understanding of pancreatic organogenesis and the establishment of novel diabetic disease models. These physiologically relevant models are great tools for screening and developing new therapeutic drug candidates for diabetes and, in extension, could be employed for subsequent patient-specific therapies. Models based on patient-derived cell sources exhibit an excellent opportunity to optimize the efficacy and specificity of novel therapeutic molecules in diabetic patients. Currently, 3D organoid models have been employed to generate β-cells for T1DM, which is aimed to be used for cell replacement therapies without the requirement of transformed and donor cells [81]. Notably, Petersen et al. have described a method to generate glucagon-producing α-cells, which are critically involved in elevating blood glucose levels through the regulation of glycogenolysis, and could benefit in understanding the disease progression in diabetes [82]. Indeed, the researchers have observed that the stem-cell-derived α-cells shared a similar structure to cadaver-derived α-cells, lack insulin expression, and secreted glucagon in response to glucose. Upon transplantation to mice, these cells showed elevated blood glucose levels in the study animals [82]. Of interest, Wimmer et al. have fabricated human blood vessel organoids from stem cells to closely mimic the microvascular alterations in diabetic vasculopathy [83]. Notably, when the blood vessel organoids were transplanted into mice, the formation of a highly stable and perfused vascular network complete with arteries, arterioles, and venules was observed. Moreover, exposure of blood vessel organoids to hyperglycemic and inflammatory stimuli precipitated vascular basement membrane thickening and other microvascular pathologies observed in diabetes [83].

It should be noted that the function of pancreatic cells derived from stem cells would be compromised without a microvasculature environment. Encompassing the vasculature into the pancreatic organoid is essential to fully understand diabetic pathophysiology, especially in microvascular complications. In this regard, a demonstration by Wimmer et al. employed organoid technology to investigate the pathophysiology of vascular complications of diabetes [83]. The researchers established a human diabetic vasculopathy model by differentiating human iPSCs into organoids connected with endothelial cells (ECs), pericytes, mesenchymal stem-like cells, and hematopoietic cells through a lumenized 3D capillary network. Exposure to high glucose and pro-inflammatory cues, including cytokines, mimics the pathophysiology of diabetes [83].

Additionally, markedly thick vascular basement membranes with a low capillary density in the dermal microvasculature of T2DM patients were also observed in the 3D diabetic organoids. Highlighting the successful applications that organoids hold, NOTCH3 and DLL4 were identified as signalling mediators of vascular pathology [26,83]. Another method to obtain vascularized organoids would be to co-culture native pancreatic tissue fragments and iPSC spheroids with human umbilical vein endothelial cells in Matrigel. Intriguingly, incorporating amniotic epithelial cells with human islet cells in a microvasculature environment improved the engraftment, viability, and function of the organoid in the T1DM mouse model [84]. Functional mini-organs engineered through such strategies have immense potential to become a promising source of insulin-producing cells.

The integration of biosensors in in vitro tissue systems has shown considerable promise in the effective measurement of glucose uptake. Obregon et al. demonstrated the integration of a nanobiosenor in a biomimetic skeletal muscle tissue model [85]. They utilized nano-porous gold particles to fabricate a highly sensitive non-enzymatic biosensor to detect the glucose uptake by skeletal muscle tissues. Notably, upon electrical stimulation of the skeletal muscle tissue model, the investigators were able to identify that glucose consumption doubled in electrically stimulated tissue compared to non-stimulated tissue [85].

Further development in organoid cultures is warranted to improve the response toward glucose stimulation and better mimic the cell-to-cell interactions. Since the organoids are obtained using stem cells, which can differentiate into many cell types, directing them towards a few particular cell types needs specific conditions and growth factors. Incomplete differentiation of these stem cells would also be problematic. Even though human islets are considered the gold standard for understanding the physiological functioning of the pancreas, the limited supply of these cells, high time consumption, and isolation expense remain significant limitations in using them. There is scope for improvement in mimicking the function, heterogeneity, and vascularity of the native system [86]. With the current advancement in gene-editing technologies, the regulatory networks and the pathogenesis can be understood to some extent. Multi-organoid systems embedded in a system known as an organ on chip (OOC) can be used to study the function/dysfunction of the pancreas in the context of signals from its neighbouring organs, such as the liver.

### 4.3. Organ on Chip

Organ-on-chip (OOC) models are microfluidic systems with well-defined patterns, scaffolds, or structures, delivering dynamic biochemical signals based on biomimetic flow. Through these chips, the ability to obtain real-time interpretations, scalability, and capacity to construct complex microenvironments with better spatiotemporal precision is possible. OOC are made using various micro-fabrication techniques, including soft lithography, photolithography, and contact printing. Upcoming evidence from studies employing OOC in preclinical drug testing and organ dysfunction has only accelerated the need to replace animal models. Aggregation and differentiation of β-cell organoids that were found responsive to glucose can be performed using multilayer OOC devices (Figure 3). Indeed, pancreas-on-chip is a rapidly advancing platform in diabetic research, which allows cell manipulation to achieve cell polarization, direct cell–cell interaction, and propagation of chemical and electrical signalling [87].

The OOC technology goes a step ahead and allows it to combine multiple cell types in distinct compartments interconnected through microfluidic channels. Since it can be considered that diabetes is caused by a miscommunication or disruption of crosstalk between different organs, OOCs can effectively understand the systemic interplay between organ systems (Figure 3). Notably, the establishment of the vasculature network is directly correlated with the investigative efficiency of this device. Attempts have been made to regulate the circulating glucose content of the insulin produced by the β-cells by combining hepatocytes and islet cells in a multiple-organs-on-chip device. Insulin stimulates the processes of glycogenolysis and gluconeogenesis in the liver and is thereby involved in the synthesis of protein and lipids. Essaouiba and Okitsu et al. have developed a pancreas–liver OOC to study insulin and glucose regulation [88]. They fabricated a peristaltic micropump on-chip by coupling human pancreatic islet micro-tissue and liver spheroids comprising primary human stellate cells and HepaRG cells. Notably, throughout the experiment, the secretory capacity of the β-cells in response to stimulation with glucose was maintained in the pancreas–liver OOC. The researchers achieved a constant pulsatile flow, which facilitated the high tissue perfusion rates, allowing the co-culture to be maintained for 15 days [88]. Aggregation and differentiation of β-cell organoids that were found responsive to glucose can be achieved using multilayer OOC devices. Recently a model was developed to study the immune-metabolic reactions that involves the co-culture of adipocytes with immune cells. Limited throughput was a limitation observed in these models, and there is scope for refinement. Hanging-drop-based islet perfusion systems based on microfluidics have been developed to study insulin release from pancreatic cells [87]. However, maintaining a constant drop volume under specific conditions was an observed challenge.

The design of OOCs varies based on the use of single- or multi-organ systems on a chip, both of which have two phases in their formation. In the preparatory phase, the biological components are introduced into microwells and given time to settle into the device. In multi-OOC devices, the single organ compartments are cultured separately with separate mediums. The second phase is associated with the physiological maturation of the system. In the case of multiple OOC devices, once all biological components have reached the required level of growth, they can be interconnected to form numerous OOC devices by switching to a common culture medium in the experimental phase [89,90]. The trapping sites in the microfluidic chips are used for culturing the cells. Due to its biocompatibility, optical transparency for better imaging applications, and easy assembly, PDMS, a silicon-based organic polymer, is the most widely used for pancreas-on-chip devices [91,92]. Transparent microfluidic systems will facilitate the seamless integration of optical detection to evaluate different biochemical and molecular readouts online [93].

A series of studies from Kennedy et al. were the first to demonstrate and evaluate the application of microfluidic devices with islet cells to monitor the functionality of β-cells [93,94,95]. The initial prototype chip was designed to constantly monitor islet cell secretions from multiple independent living samples. The insulin secretion from islets of Langerhans was used to evaluate the performance of the device. The chip was equipped with four distinct channel networks containing islets, each capable of performing electrophoresis-based immunoassays of the perfusate. However, the lack of perfusion in the islet chamber was a limitation in dynamic insulin secretion and affected the viability of the islet cells [94]. Advancements in this system incorporating continuous perfusion and online electrophoresis in the islet compartment showed temporal resolution for monitoring insulin secretion while enhancing compatibility with longer-term measurements [95].

Rocheleau et al. fabricated a microfluidic device that stimulates an islet while facilitating the observation of the NAD(P)H and [Ca^2+^]i responses. By trapping the islet cells and holding them between two different glucose concentrations, they demonstrated that β-cells could be effectively coupled to coordinate an intracellular Ca^2+^ response only within regions of the islet in contact with a glucose concentration above the physiological threshold [96]. However, the β-cells could not be coupled properly below the threshold glucose concentration. This model emulates the delicate interaction between the degree of coupling, the extent of ATP-sensitive K^+^-channel activation, and the role it plays in Ca^2+^ propagation across the islet. Islets on chips have also been established to serve the purpose of measuring insulin via glucose pulses. Chen et al. developed a plug-based microfluidic device termed a “chemistrode” to stimulate, record, and analyze molecular signals from a single murine islet. This device exhibited high temporal, spatial, and chemical resolution for stimulation and recording, with pulses captured as short as 50 ms [97]. In line with this evidence, Misun et al. demonstrated a method to study the dynamics of insulin release of single pancreatic micro-tissues at a high temporal resolution. They fabricated a microfluidic hanging-drop-based perfusion system that enabled rapid glucose switching, minimal sample dilution, low dispersion of the analyte, and reduced sampling intervals. This model can potentially be used to study physiologically relevant dynamics in insulin secretion with low sample-to-sample variation and high temporal resolution [98].

The number of studies employing pancreas-on-chip for the physiological analysis of pancreatic islets is still limited compared to other organs or tissues. OOCs for diabetic research are still in the early stage of advancement, and there are limitations in several physiological aspects that have to be resolved. Pancreatic islets are micro-organs exhibiting high vascularization through a dense network of capillaries to enable a seamless exchange of nutrients, oxygen, and hormones between the systemic circulation and endocrine cells. Hence, it should be noted that the disruption of the islet cell vasculature reported in islet cell culture is of major concern. Indeed, this could be the underlying reason for the lack of long-term sustainability of primary islets [99].

Further, the current pancreas-on-chip models do not consider the exocrine part of the organ surrounding the islet cells. In addition, the heterogeneity in the size of the islet cells produced through the microfluidics platform poses a problem. In terms of multiple OOC devices, the specificities of each of these organs need to be evaluated to construct a chip mimicking the microenvironments at the ratio of the in vivo organ tissues. Moreover, developing a universal culture medium in microfluidics for the co-culture of cells could be beneficial.

## 5. 3D Scaffold-Free System

The employment of scaffold-free techniques facilitates the self-assembly of cells into non-adherent aggregates referred to as spheroids. However, it should be noted that the spheroids can also be incorporated into several scaffold systems. The scaffold-free cultures address some drawbacks of scaffold-based cultures, including biocompatibility, that arise due to the use of non-human materials employed in fabricating the scaffolds [100]. In a scaffold-free system, the spheroids simulate the solid tissues through the secretion of their own ECM and display differential nutrient availability. Moreover, the spheroids grown via non-scaffold-based techniques are characterized by their consistency in size and shape. They are observed to exhibit a better performance as in vitro models for high-throughput screening. Scaffold-free systems have evolved as pioneering strategies for generating in vitro components that exhibit high cellular viability and the ability to mimic the biological environment [101]. Since spheroids are comprised of human cells and their secreted ECM only, scaffold-free approaches are well-suited for bio-analytical studies aiming to address fundamental scientific questions such as protein abundance regulation due to a 3D environment. In these scenarios, the lack or absence of a protein-sourced scaffold would particularly aid in the analysis of ECM deposition and protein secretion through mass spectrometry (MS)-based proteomics [100]. Several scaffold-free systems have been employed to develop neo-tissues complete in functionality that are applicable for clinical targets [102]. Vascularized pancreatic β-islets have been bioengineered by anchoring the cells to PhenoDrive-Y, a biomaterial based on gelatin, which facilitated mimicking the basement membrane of tissues [103]. Such models have the potential to study the functional relationship between angiogenesis and islet formation in diabetes. Considering that the dysfunction of adipose tissue is crucial in the development of type 2 DM, scaffold-free adipose spheroids have been generated from pre-adipocytes and exhibit a better response to stress, greater adiponectin and pro-inflammatory cytokine secretion, and robust retention of brown adipose tissue (BAT) markers compared to 2D culture systems [104]. Certain organoids have also been developed through scaffold-free techniques. Scavuzzo et al. demonstrated pancreatoids, which were obtained by the self-assembly of β-cells in scaffold-free conditions and the presence of native mesenchyme [105]. They were reported to exhibit properties of insulin-secreting endocrine cells. Spontaneous self-aggregation, hanging-drop plates, and suspension cultures are some commonly used methods to generate scaffold-free systems. Some of the most frequently used applications of scaffold-free systems include the employment of non-adhesive wells, rotating wall vessels, magnetic levitation, and bioprinting [100].

### 5.1. Spheroids

Single cells in suspension clump together to create a loosely bound cluster of cell aggregates in spheroids. Further, the formation of extracellular matrix with cell surface compacts the cell aggregates to form sphere-like structures [106]. Spheroids are formed during the proliferative phase and provide the distinct advantage of being able to control the size of spheroids and simulate organ-like micro-architecture and morphology [107]. Broadly, the formation of spheroids comprises three distinct stages: (i) the formation of loose cell aggregates via integrin–ECM binding; (ii) a delay period for cadherin expression and accumulation; and (iii) the formation of compact spheroids through homophilic cadherin–cadherin interactions (Figure 4) [108]. The interactions between cells in spheroids and the ECM have been observed to mimic the in vivo microenvironment closely [109]. In this regard, multicellular spheroids could be employed to create more complex tools in diabetic research [110]. Several sources of cells, including stem cells, islet cells, and adipose cells, have been employed to create study models of spheroids for understanding pancreatic function. In some spheroid models, the co-culture of these cells is used to study the intercellular crosstalk [111].

Four major methods have been used to generate spheroids: self-aggregation, hanging drop, suspension culture, and scaffolding (Figure 4). Spontaneous self-aggregation generates spheroids through the seeding of cells on ultra-low-adhesion plates. These specialized plates enable the achievement of spheroids with a defined geometry and facilitate high-throughput screening throughout the workflow. However, it should be noted that this spontaneous self-aggregation method cannot control the number of cells per spheroid. The hanging-drop technique has been used to generate multicellular or co-culture spheroids. Cells in the media are initially dispensed on the top of the hanging-drop plate (HDP) well, which then allows the cells to sort out into discrete droplets generated below the HDP well. However, this method requires technical skill to transfer the spheroids from the HDP well to a second plate for assays. The suspension culture method employs bioreactors, which drive the self-aggregation process of cells into spheroids under dynamic and highly controllable conditions. This method exhibits the added advantage of enabling the large-scale production of spheroids. Still, this method is also associated with a few disadvantages, including non-specific spatial organization and low throughput.

Using a 3D clinostat microgravity generator, large quantities of spheroids were generated using MIN6 pancreatic cells. The simulated microgravity culture system enabled the spheroids to be produced at the required size based on the cell density used. Various β-cell signature genes were expressed at an elevated level in the spheroids compared to a conventional 2D culture dish. More importantly, when these spheroids were transplanted into the portal vein of streptozotocin (STZ)-treated diabetic mice, it ameliorated hyperglycemia. In contrast, the equivalent transplantation of 2D-cultured cells could not show a similar effect in the study animals [112]. Spinner culture methods have been used to promote spheroid formation by keeping cells in suspension through agitation or by increasing the viscosity of the media, allowing spontaneous aggregation [108]. However, high-throughput suspension cultures lack size and precise environmental homogeneity control [52,108]. Spheroids can also be generated using various micro- or nano-patterned scaffolds. They have very few plate-to-plate differences, so high-throughput screening is possible. However, the formation of bubbles in these cultures is a frequent problem, and removing them could cause damage to the micro-patterned surfaces [59]. Pancreatic islet spheroids employed for 3D cultures and transplantation have been demonstrated using self-aggregation, hanging-drop, and micro-well systems [113]. Wassmer et al. compared the efficiency of various spheroid-generation methods with native islets. They observed that the self-aggregated spheroids were more prominent and exhibited a different size and shape when compared to spheroids generated through other methods [113].

Interestingly, spheroids generated using the hanging-drop method, Sphericalplate 5D™, and agarose microwell plates exhibited a uniform and well-defined round shape. However, self-aggregated spheroids had an improved insulin secretion response on glucose stimulation compared to native islets. Thus, different techniques pose distinct advantages and inconveniences, which warrant further refinement to achieve reproducible, large-scale spheroid production [113]. In this line, Lee and Hong et al. have established a microphysiological analysis platform (MAP) to facilitate the uniform 3D spheroid formation of pancreatic β-cell islets [110]. The MAP platform also allows for morphological phenotyping at a larger scale and mapping of gene expression of chronic hyperglycemia and hyperlipidemia in DM. Densely packed β-cell spheroids were generated in a scaffold-free formation equipped with a perfusion flow network to mimic the in vivo microenvironment. It has also been reported that disease conditions such as glycemia, lipidemia, and other dynamic perturbations were precisely controlled due to the MAP system, which enabled the researchers to study the role of hyperglycemia and hyperlipidemia on β-cell apoptosis [110]. Essaouiba et al. compared the efficiency of 2D biochip culture with 3D spheroids generated from honeycomb static cultures [114]. They observed that when compared to 2D cultures, spheroids in static cultures demonstrated superior expression of β-cell and α-cell markers and exhibited higher insulin secretion ability in response to high/low stimulation with glucose. The researchers went on to inoculate these spheroids into biochips, which were maintained in perfusion for 10 days. They reported that the 3D biochip cultures exhibited improvement in the expression of β-cell marker signature and responsiveness to glucose. These results highlight the potential of improving a static spheroid culture by incorporating it into a 3D chip model for disease modelling and anti-diabetic drug screening [114]. In this regard, a microfluidic two-organ chip model has been established to investigate the role of complex crosstalk between pancreatic islet and liver in regulating insulin and glucose. Bauer et al. have co-cultured human pancreatic islet micro-tissues and liver spheroids to establish a ‘functional coupling’, represented by glucose-stimulated insulin secretion from the islet micro-tissues and consequent glucose uptake by the liver spheroids [115]. In stark contrast to single-cell cultures, the postprandial glucose concentrations were maintained in the circulation of co-cultures. The liver spheroids did not consume glucose efficiently in the absence of insulin. With reproducibility, the researchers thereby demonstrated a functional feedback loop between the liver and the insulin-secreting islet micro-tissues [115]. Even though various spheroid generation methods have been demonstrated, significant challenges still exist, which include the lack of full ability to control the spheroid size, reproducibility, effectiveness of spheroid formation, and complex liquid handling methods [110]. DIPCs (differentiated insulin-producing cells) have evolved as a novel source of islet cells for therapeutic purposes. Their application in natural islet formations could enhance spheroid structure and differentiation [116]. Feng et al. have developed a non-cross-linked hyaluronic-acid gel that contains therapeutic spheroids derived from human ASCs (adipose-derived stem cells) [77]. On injecting these spheroids into diabetic rodents with a delayed wound-healing model, they observed a markedly elevated rate of healing coupled with angiogenic potential at the site of diabetic ulcer [77]. Such ready-to-inject micro-spheroids have opened new avenues in improving the therapeutic efficacy in treating impaired wound repair. The MAP-based human diabetes models and 3D multi-organ chip micro-spheroid models could also pose an efficient surrogate to existing animal models or cell lines derived from rodents [77].

### 5.2. Bioreactors

Specialized bioreactor systems such as micro-bioreactors (MBRs) offer incredible potential for growing mammalian tissues and cells under physiological conditions. Unlike the classical bioreactors, mainly used for industrial-scale production, MBRs provide the opportunity to use small quantities of chemical entities and a low number of cells, especially when the primary cell/tissue is of limited availability. These MBRs are considered an intermediate stage that could facilitate the development of more complex OOC systems [117]. Bioreactors can achieve effective cell expansion by assessing a significant number of factors such as substrate consumption, metabolite synthesis, cell proliferation, and differentiation. Bioreactors allow for high-precision control of critical parameters, including pH, temperature, active fluid flow, and gas supply [48].

Moreover, the bioreactors employed in 3D cell culture can be designed to allow a frequent analysis of the culture through techniques such as microscopic imaging during their operation. Compared to cells grown in two-dimensional static cultures, cells grown in bioreactors exhibit much higher levels of division, greater transcriptional communication, and improved translation of pancreatic-tissue-specific genes. Bioreactors represent the exciting possibility of constructing channels with external openings using 3D culture, making it feasible to regulate and maintain the human islets in a physiological environment. These channels would allow the transportation of nutrients and oxygen into the islets after they have been transplanted, improving the functionality of human islets [48].

Growing MSCs in bioreactors requires a high surface-to-volume ratio, a closed system, automated inoculation and harvesting, and online parameter control. Human MSCs can be grown in reactors by employing traditional fixed beds, fluidized beds, stirred tanks, wave reactors, wall-rotating systems, and vertical-wheel components [52,118]. The mass production of β-cells is coupled with inherent challenges, particularly glaring in traditional 2D cultures, where β-cells have been observed to lose their functionality. Three-dimensional culture systems such as bioreactors offer a means to preserve the β-cell phenotype and improve the regulation of insulin gene expression. The major goal of such a system would be to attain a high percentage of β-cell viability and to increase its glucose-stimulated insulin-secretion response. Rotating-wall vessel reactors (RWVR) have been utilized to aggregate and cultivate human and mammalian islets [118] (Figure 5). Even though RWVRs apply low shear stress, it provides the advantage of appropriate mixing, which ensures nutrition and oxygen transport to cells. The continual rotation of the medium creates microgravity, which suspends the islets. Notably, the murine islets cultivated through bioreactors exhibited improved functionality, as indicated by higher SI values than freshly isolated cells [118].

Moreover, multiple nutritional channels were developed in the islets cultivated from bioreactors, with robust insulin gene expression and response to glucose stimulation [117]. Furthermore, human islets cultivated in an RWVR exhibited stable islet structure and improved functionality compared to 2D static cultures. RWVRs have also been utilized to create hepatic tissue architectures from embryonic liver cells, demonstrating the usefulness of this technique for the reconstruction of endodermal organs, such as the liver and pancreas. RWVRs, however, suffer from a drawback due to the spontaneous clumping of islets.

Stirred-tank reactors (STRs) and fixed-bed reactors (FBRs) have also been utilized to cultivate β-cells (Figure 5). Human pancreatic islet cells were trypsin-dispersed for growth in stirred-tank reactors. β-cells have been cultivated in aggregates or capsules, in contrast to human MSCs, which are expanded in microcarriers. However, β-cells that were expanded in microcarriers such as Cytodex-3 in an STR system produced close to 2.6 times more insulin when compared to 2D culture systems [119]. FBR systems equipped with alginate-filled hollow fibers have been employed to culture porcine pancreatic cells to generate cell aggregates. Notably, these cells exhibited excellent viability and synthesized more insulin than aggregate suspension culture [52].

Adipose tissue is a vital endocrine organ, and the long-term development of human adipocytes in vitro would provide metabolically active tissues that could serve as diagnostic models to screen therapeutics targeting diabetes. The introduction of dynamic 3D perfusion to a bioreactor system has been reported to facilitate mature adipocyte generation and provide a source for adipose tissue for 2 months. More importantly, these multicompartment 3D-perfusion bioreactors exhibited reproducible metabolic activity [52] (Figure 5). Semi-permeable hollow fibers in a multicompartment architecture allow uniform nutrition and gas exchange with greater physiological gradients, integrated oxygenation, and pH control to the total cell compartment volume with minimal shear stresses [120] (Figure 5). Spinner flasks are not appropriate for the large-scale growth of cells since they lack the option to control the environment [52,121]. The long-term culture of non-mechanical cells is more accessible in these bioreactors.

Moreover, the incorporation of microgravity, another essential feature in bioreactors, has been successfully implemented to improve the functionality of pancreatic islets. Microgravity reactors enable a continuous circular rotation of the medium and allow for the growing cells to be suspended in continuous free fall at a terminal velocity with the benefits of low hydrodynamic shear stress force and minimal turbulence along with high mass and oxygen transfer. This reactor structure has also been implemented successfully in islets obtained from human patients with hyperinsulinemia hypoglycemia of infancy (PHHI). It has produced an environment suitable for elevated endocrine expression [121,122]. The propensity of cultured islets to aggregate and lead to anoxic necrosis when cultivated at higher densities necessary for successful human islet transplantation is the primary disadvantage of islet culture inside these bioreactors [121]. Further experimentation and refinements to these techniques, such as incorporating solid supports, including microcapsules and scaffolds, are warranted to enhance the scale of islet culturing.

The shortage of allogeneic donors remains a major limiting factor for β-cell replacement therapy and for obtaining source cells in pancreatic 3D models. Prenatal and adult pigs exhibit a good prospect for obtaining cells for islet xenotransplantation [123]. Due to β-cell expansion, neonatal pig islet cells have a more excellent functional capability to rehabilitate to normoglycemic conditions [124,125]. In the case of an adult pig, a large number of viable cells can be obtained for xenografting, with the advantage of these cells being fully mature and functional immediately after implantation. However, immune rejection remains a major bottleneck in the clinical application of these cells. Immune-protected encapsulated islets overcome this issue and suffer from a low oxygen supply. In this regard, the generation of low-immunogenic pig pancreatic islet cell clusters (ICC) for xenotransplantation promises a high clinical utility [126]. Bioreactors offer great potential in generating low-immunogenic porcine ICCs, which could be an alternative to allogeneic pancreatic islet cell transplantation. Porcine pancreatic ICCs were enzymatically digested to obtain single-cell suspensions of islets. To confer immunogenic protection to pancreatic islets, the swine leucocyte antigen (SLA) class I and class II were silenced using lentiviral vectors that encode for short hairpin RNAs (shRNAs), which target β-2-microglobulin or class II transactivator. Stirred bioreactors were employed to obtain low-immunogenic ICCs by growing SLA-silenced ICCs in the presence of collagen VI. Notably, T-cell-, antibody-, and NK-cell-mediated immune responses and xenogeneic T-cell immune responses were markedly reduced in SLA-silenced cells. Moreover, the tissue-engineered islets exhibited a characteristic 3D structure with efficient insulin production in stirred bioreactors [126].

### 5.3. 3D Bioprinting

3D bioprinting is used to fabricate biocompatible materials, cells, and supporting components to create functioning living tissues. This technique offers incredible potential for usage in regenerative medicine to meet the growing need for transplantable tissues and organs. Compared to non-biological printing, 3D bioprinting has more special considerations about selecting the material, compatible cells, and growth and differentiation factors, as well as accounting for the specific sensitivities of tissue formation [68]. Three-dimensional bioprinting uses additive manufacturing technology to print cells, biomaterials, and cell-laden biomaterials layer by layer, resulting in 3D tissue-like structures. It is an efficient and economical platform that allows for the exact spatial positioning of cells, proteins, genes, medicines, and biologically active particles to better control tissue creation and growth. Three-dimensional bioprinting is the most commonly used method for fabricating artificial tissues such as blood vessels, heart, bones, liver, and skin.

The process of bioprinting to generate tissue constructs consists of three key stages, which include pre-bioprinting, bioprinting, and post-bioprinting (Figure 6). Pre-bioprinting comprises medical imaging, cell culture and expansion, 3D modelling or designing, and G-code generation. This is followed by the bioprinting stage, which involves bio-ink preparation, printing, adjustment of cellular parameters, and 3D construction. The final stage, post-bioprinting, comprises 3D construct maturation, nutrient and oxygen supply, and mechanical and biochemical support [127,128]. Bioprinting can be broadly classified based on biomimetics, autonomous self-assembly, and mini-tissue building blocks. Biomimetic components included in a bioprinted construct have a dynamic influence on the adhesion, migration, proliferation, and function of endogenous and foreign cells. To achieve the micro-scale replication of biological tissues, a knowledge of the microenvironment, including the cellular makeup, soluble and insoluble substances, and the ECM, is essential [68]. The autonomous self-assembly method seeks to mimic embryonic environmental and structural components to create accurate embryologic architecture. This method could be utilized to develop micro-tissues from isolated components that can self-organize [53]. Subsequently, accurate, high-resolution copies of a tissue unit would be developed and allowed to self-assemble into a working macro-tissue. Self-assembly of vascular building blocks to generate branching vascular networks has also been achieved through applying these methods [54,129].

The three most commonly used techniques employed in bioprinting include: (i) inkjet-based bioprinting, (ii) laser-assisted bioprinting, and (iii) extrusion-based printing (Figure 6). Inkjet-based printing modules may deposit cells or biomaterials as droplets using heating reservoirs or piezoelectric actuators. The printing nozzle’s heating element elevates the temperature, causing gasification and bubbles, subsequently printing bubble droplets onto a substrate. Piezoelectric inkjet bioprinters produce cell-containing droplets via the nozzle [130,131,132]. An inkjet printer may be used to position the cells precisely in predetermined patterns through a technique referred to as cytoscribing, where cell-adhesion proteins are deposited on a substratum through computer control. Laser-assisted bioprinting employs an infrared source to evaporate biological components that are coated onto a ribbon, which are subsequently gathered as droplets by the substrate [133] The main advantage of this method is its facilitation of the deposition of bio-inks with markedly higher viscosity and resolution. Bioprinting based on extrusion employs pneumatic or mechanical dispensing methods to extrude continuous cell beads. In addition, it can manufacture cell-laden bio-inks in the form of continuously extruded strands capable of creating large-scale biomimetic structures due to their rapid printing speed [134,135].

Even though 3D bioprinting has been successfully implicated in the fabrication of tissues such as blood vessels, skin, liver, bone, and cartilage, the studies reporting bioprinting pancreatic islet tissues for diabetic research are limited. Stem cells employed with 3D printing have great potential for strategizing treatments for regenerating the damaged pancreas and in diabetic drug delivery [134]. Under the pathological conditions of hyperlipidemia and hyperglycemia, there is considerable damage to the vasculature, resulting in micro- and macrovascular complications. The replication of the in vivo vascular-like structure is a requirement that can be met through bioprinting [136,137]. In particular, extrusion-based bioprinting has great promise for mimicking structures in vascular diseases. Coaxial printing mimics the vascular architecture in vitro by placing the endothelium layer in the core with sacrificial inks and the smooth muscle layer in the shell to form a 3D tubular structure [26]. While such sacrificial inks have been mainly employed for void-space creation and printing of pattern perfusable networks, they also offer the potential to be directly combined with the bio-ink to modify its mechanical properties, improve printability, and elevate porosity. Interestingly, scaffold-free tubular tissues that were fabricated using bioprinting underwent remodelling and endothelialisation after implantation in rat aortae [138]. Alginate-based bio-ink has been employed to print human islets and rodent pancreatic cells into a predetermined 3D scaffold [139]. Extra-hepatic islet delivery systems were developed by Marchioli et al. using alginate-based porous scaffolds through 3D plotting. The 3D-plotted constructs were then utilized to embed INS1E β-cells and human and mouse islets without altering the cellular morphology and viability while attenuating their aggregation. Notably, it maintained the surface-to-volume ratio and therefore, the oxygen and nutrient transport in 3D-plotted alginate scaffolds were increased compared to conventional bulk hydrogels. Moreover, the use of alginate/gelatin mixtures led to an improvement in the plotting performance and handling properties [140]. Studies indicating the promise of treating diabetes by incorporating bioprinting technology need further experimentation.

When bioprinting multiple types of pancreatic cells into a tissue construct, engineered islets must be fabricated, followed by encapsulation in the form of a spheroid, and deposited into a vascularized tissue construct. This allows the engineered islets to self-assemble into a highly functional islet structure. These islets can then be built in three dimensions, employing various biofabrication techniques. Engineered islets may be prevascularized to facilitate capillary development, followed by bioprinting of the gel and encapsulation to replicate natural pancreatic endocrine activity [134]. Using a multi-head, deposition-based 3D-printing system and alginate bio-ink, an insulin-secreting implantable construct that usually functions for an extended period of time by expanding into large cell aggregates with a diameter of 100–200 m has been reported [141]. Akkouch et al. took advantage of tissue self-assembly and guided the fusion of tissue strands to facilitate the fabrication of larger tissue patches. They demonstrated the fabrication of scaffold-free tissue strands comprising rat fibroblasts in the core, and mouse insulinoma TC-3 β cells in the shell, by employing tubular alginate conduits as mini-capsules with defined permeability and mechanical properties [142].

Bioprinting can recreate complex morphologies and multicellular systems, allowing for the production of pancreatic-tissue-like structures. This method also overcomes the drawbacks of conventional islet encapsulation technologies, such as hypoxia and lack of vascularization [127]. Three-dimensional bioprinting technology with increased precision might offer highly regulated seeding of islets, limiting the development of pericapsular fibrotic overgrowth (PFO), which causes islet necrosis. The cell viability and morphology were found to be unaffected in the printed cells [134]. Bioprinting with pancreatic islets followed by transplantation into diabetic mice led to controlled insulin secretion [138,143]. However, it should be noted that not all cell types are compatible with a certain bioprinting method [144]. Indeed, certain unsuitable bioprinting materials may lack the gelling and mechanical qualities that support islet function. It should be noted that to create a working β-cell replacement therapy (BCRT) transplantation model, bio-inks that are compatible with both β-cells and islets with the ability to print need to be chosen. Moreover, protection from autogenic and allogenic or xenogeneic immunological reactions to the graft/product should be considered [144]. Three-dimensional bioprinting technology may improve islet cell survival and function by generating a 3D structure akin to human tissue, utilizing β-cell bio-ink. Optimizing the bio-ink composition has the potential to overcome the limitations of using simple islet transplantation or subcutaneous implant patches [141]. Considering the challenge of attaining high printability and biocompatibility using synthetic bio-ink, further focus on developing naturally derived bio-ink is warranted.

The integration of biosensors into 3D systems could further improve the utility of these models by providing new perspectives from previously unavailable data. In particular, the monitoring of interstitial glucose and glucose uptake in a non-invasive manner using an integrated biosensor has great value in understanding the effectiveness and validity of a diabetic 3D model. Liu et al. demonstrated the use of a microneedle biosensor obtained through a 3D bioprinting process. The development of the device consisted of obtaining the microneedle array through 3D printing, followed by microfabrication and electroplating steps to arrange the electrochemical sensing electrodes on the microneedles. The fabrication process was concluded after enzyme immobilization with glucose oxidase (GOD) on the working electrodes of the biosensor. Integration of this device into the dermis layer of rodent skin facilitated the accurate and continuous measurement of glucose levels [145].

## 6. Future Prospects and Improvement in 3D Models

Recent evidence persuasively supports the arguments for applying and adapting appropriate 3D cell-culture technology to diabetes research. Since the onset of 3D culture technology, high-fidelity models of various body tissues have been generated, which offer markedly higher predictive power over conventional 2D models. Further refinements of the 3D models could considerably increase the reliability and reproducibility of drug discovery while also enabling the identification of novel disease pathways implicated in diabetes. Especially for a disease such as diabetes that affects multiple organ systems, where the inter-organ crosstalk is of particular importance, OOC technology has made massive strides in incorporating various organ systems to document their involvement and response to altered signalling mechanisms in DM. However, the challenge is to effectively combine the OOC technology with organoid technology and incorporate a continuous perfusion strategy. Chip-based technologies perfused with pumps have already been developed and offer considerable potential to be employed in upcoming studies.

Further research is warranted to include human organoid tissue models using OOC-based technology to study systemic communication between organs in diabetes. Insulin-producing cells have exhibited increased viability and insulin secretion when cultivated in 3D cultures. Organoids can be used as an alternative to pancreas organ transplantation and organ grafts to overcome the lack of high-fidelity cells and limited maturation issues. Three-dimensional bioprinting has already made a significant contribution to medicine and medical devices by allowing for the creation of customized and personalized dosage forms, implants, and other goods for patient compliance and adherence. In diabetes research, 3D bioprinting offers an excellent opportunity for application in regenerative medicine. However, the 3D bioprinting models must be scaled up to dimensions relevant to human clinical use.

The potential of 3D models in being able to closely replicate complex in vivo cellular morphology, tissue architecture, surface chemistry, biomechanical properties, and spatiotemporal relationships coupled with difficulty in the systematic assessment and validation of the models should be noted. Most models have not yet reached the preclinical phase owing to strategy-specific limitations encountered by these models [59]. Compared to traditional 2D cultures, the analytic assays available for 3D models are limited.

Although stem cells could differentiate into functional cells in vitro, not all cells complete differentiation, and a fraction remain undifferentiated [146]. Even after incorporating a basement matrix and assistive cells, the crosstalk between neighbouring cells and tissues is not perfect [51], and the incorporation of vasculature has not been mastered. Especially in the 3D models employing primary islets/β-cells, limited availability, high cost of islet isolation, and inter-donor differences remain glaring limitations [114]. The development and integration of biochemical, electrochemical, and optical sensors that can be integrated into existing 3D systems offer a great opportunity for the non-invasive and real-time monitoring of various aspects of 3D cell behaviour including cell migration, proliferation, and viability, which are otherwise challenging to measure through conventional analytical methods. Making additional enhancements to the environmental niche by including vessel-specific ECM components might be productive in improving in vitro vascularization [26]. Mammalian islet-like aggregates can be successfully grown in suspension bioreactors using media that does not contain serum to account for the shortage of primary β-cells [147]. Organoids, which are already powerful study tools owing to their in vivo-like composition, can be further improved through microinjection or by plating the organoids onto a 2D semi-permeable membrane. Further improvements and experimentation in this technology, such as incorporating solid supports and microgravity in bioreactors, will enhance islet-culturing techniques.

## 7. Conclusions

Even though diabetes research has evolved dramatically over the last few decades, a global diabetic epidemic looms over us, with an expected explosion of diabetes-associated health complications in the coming years. The difficulty and expense of obtaining human β-cells and insufficient strategies to cultivate these cells have crippled the efforts to establish cell-based diabetic therapy. Although certain experimental drugs show high therapeutic potency in improving β-cell viability and functionality in both 2D in vitro and rodent models, researchers fail to replicate the efficacy in humans due to the inherent limitations of these study models. Three-dimensional models address the lacunae in this space by allowing for microenvironment replication in barrier tissues, thereby eliminating the dependency on animal models. Moreover, the adoption of 3D models would remove the legal and ethical constraints that are associated with the use of animals in research. Compared to 2D models, 3D diabetic disease models offer greater drug-prediction ability and inherently save valuable time, improving the translational value in drug discovery. Indeed, evidence indicates that growing β-cells under 3D conditions markedly enhances their longevity and functionality while retaining their morphology, physiological maturation, and ECM–cell interactions. Notably, 3D technologies such as OOC add the features of vascularization, perfusion, and inter-organ crosstalk to account for the interaction between organ systems in the background of hyperglycemia. The utility and sensing capacity of the diabetic 3D models can be amplified further through the integration of biosensors for the continuous and accurate measurement of previously inaccessible information on cell behaviour in 3D systems. Pancreatic organoids offer an incredible potential to be used as an alternative to pancreas organ transplantation and organ grafts in the upcoming years. Three-dimensional models that bridge existing in vitro and in vivo models while providing the needed complexity, predictivity, and biomimicry of a human physiological system would improve the future developments in diabetic research. However, the 3D systems hold some limitations. With great complexity, there are more significant challenges to overcome, warranting further improvements and experimentation in this technology before its mass adoption as the go-to research platform in diabetes and associated complications. Persistent and meticulous efforts in improving and combining features to construct new or refine existing pancreatic 3D models would open new opportunities and greatly accelerate future advancements in diabetic research.

## Figures and Tables

**Figure 1 pharmaceutics-15-00725-f001:**
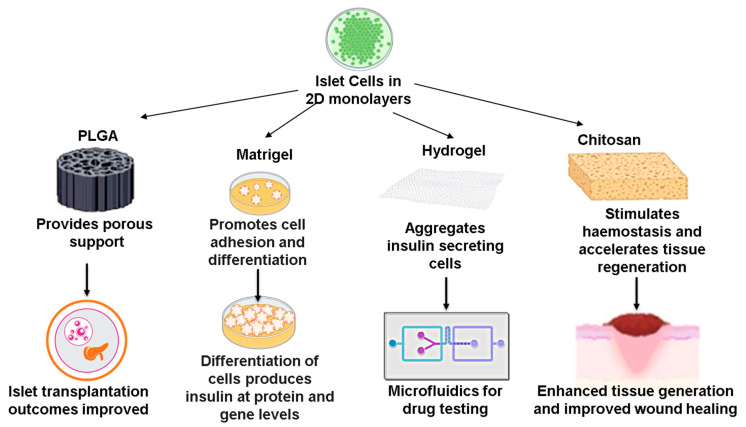
Methodologies to establish pancreatic islet cell scaffolds: Pancreatic scaffolds can be established using biopolymers arranged to mimic the ECM in the physiological state. Scaffold systems can be synthetic or biological, with porous PLGA, Matrigel, 3D hydrogel matrix, and 3D chitosan scaffolds most commonly used. Incorporating the cells into a structural scaffold-based system promotes the differentiation of stem cells into insulin-producing cells and maintains a glucose stimulation index similar to that of fresh islets. Furthermore, major desirable features such as enhanced insulin and c-peptide release, high drug-screening sensitivity, improved pancreatic transplantation outcome, and elevation in β-cell signature genes are observed in pancreatic β-cells grown on scaffold systems.

**Figure 2 pharmaceutics-15-00725-f002:**
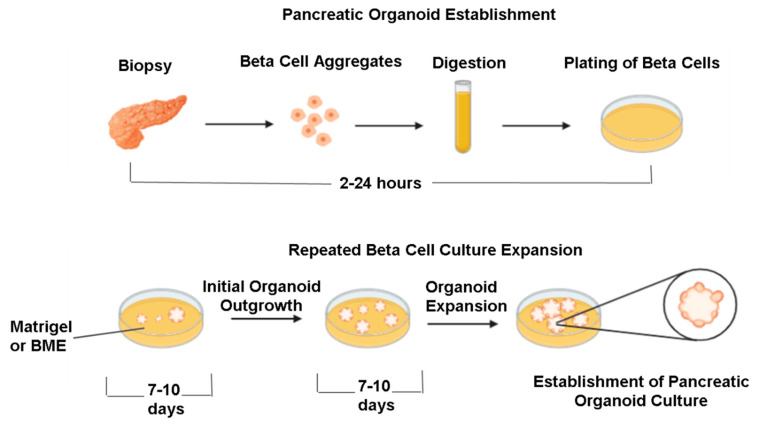
Generation of pancreatic organoids: Pancreatic organoids can be generated from primary fetal and adult pancreatic tissue or pancreatic ductal adenocarcinoma. The source of the starting tissue dictates the distinction between organoid morphology and its cellular composition. Organoids established from normal pancreatic tissue are grown on a medium containing growth factors. After the initial organoid outgrowth, they are expanded on Matrigel or BME. Normal pancreatic organoids have cystic structures and are visible 7–10 days after plating. After expansion, the pancreatic organoids can be characterized and used for downstream applications or cryopreserved for biobanking.

**Figure 3 pharmaceutics-15-00725-f003:**
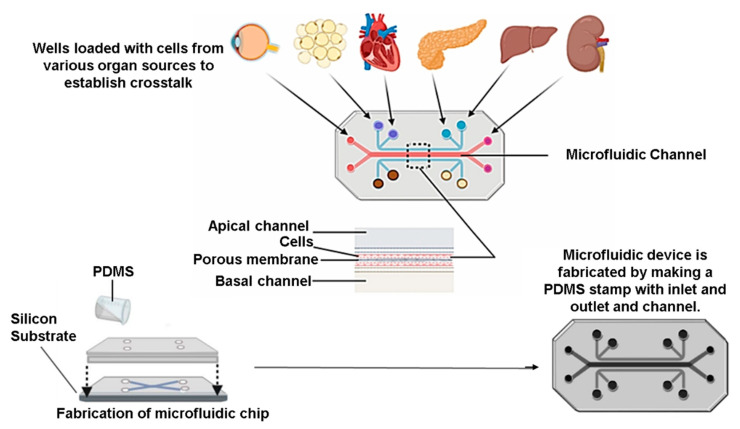
Arrangement of microfluidics system in diabetic research: To generate microfluidic culture systems, replicating patterns are etched into silicon chips in biocompatible and flexible materials. Liquid polymers, such as poly-dimethylsiloxane (PDMS), are poured onto the etched silicon substrate and allowed to polymerize into an optically transparent material to create a rubber stamp. This fabrication method is also used to develop open cavities in the form of small, linear microfluidic channels with openings at both ends of the polymer block for the perfusion of fluids. The optical clarity allowing for the real-time, high-resolution optical imaging of cellular responses to environmental cues is an advantage of the PDMS-based culture systems. Single cells are cultured as monolayers on PDMS substrate on one side of the microfluidic channel through which the medium is perfused. This system can be extended to include two or more microchannels connected by porous membranes, lined on opposite sides by different cell types, to recreate the interaction between other tissues.

**Figure 4 pharmaceutics-15-00725-f004:**
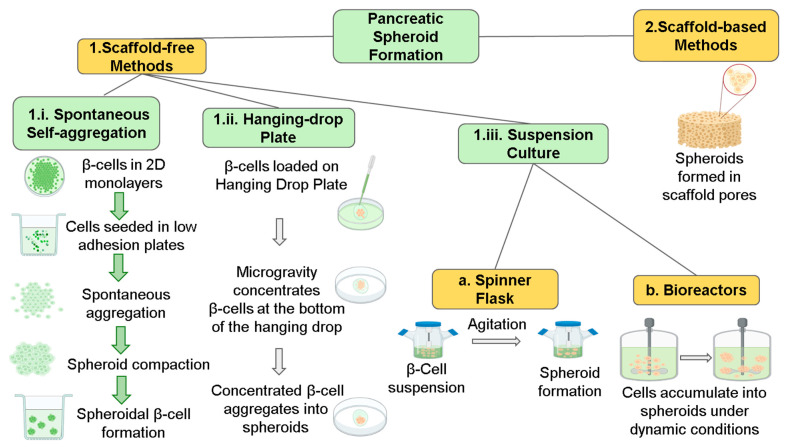
Methods to generate pancreatic 3D spheroids: Various methods to produce a 3D spheroid are adapted in diabetic research. The methods are broadly classified as scaffold-free methods, where the cells are formed as non-adherent aggregates that mimic their own solid extracellular matrix properties, and scaffold-based methods, where the cells are embedded in a matrix that influences the chemical and physical properties of the spheroids. In a spontaneous self-aggregation method, the spheroids are generated through seeding cells in ultra-low-adhesion plates that enable the achievement of spheroids with a defined geometry and facilitate high-throughput screening throughout the workflow. Spinner culture methods promote spheroid formation by keeping cells in suspension through agitation or by increasing the viscosity of the media, allowing spontaneous aggregation. The cells aggregate into droplets due to microgravity in the hanging-drop method.

**Figure 5 pharmaceutics-15-00725-f005:**
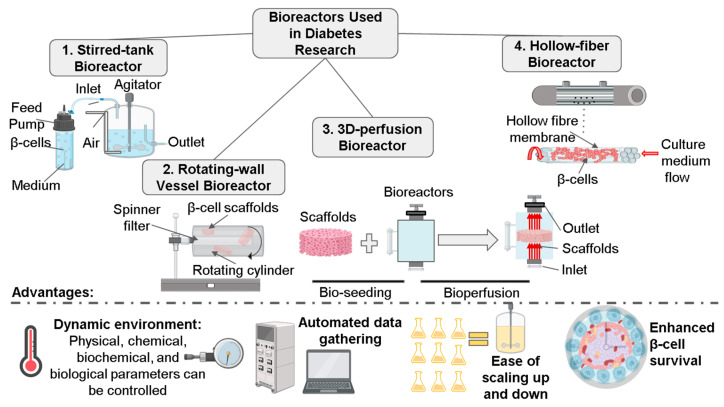
Various bioreactors in diabetic research: Bioreactors are capable of achieving effective cell expansion by assessing significant factors such as substrate consumption, metabolite synthesis, cell proliferation, and differentiation, and allow for high-precision control of critical parameters including pH, temperature, active fluid flow supply, and gas supply. Four types of bioreactors have been commonly employed including (1) stirred-tank bioreactors (2) rotating-wall vessel bioreactors (3) 3D-perfusion bioreactors, and (4) hollow-fiber bioreactors. β-cells grown in stirred tank bioreactors exhibited excellent viability and synthesized more insulin in comparison with aggregate suspension culture. The rotating-wall bioreactors provided the advantage of appropriate mixing, which ensures nutrition and oxygen transport to cells. Multicompartment 3D-perfusion bioreactors exhibited reproducible metabolic activity. Hollow-fiber bioreactors facilitate uniform nutrition and gas exchange with greater physiological gradients, integrated oxygenation, and pH control to the total cell compartment volume with minimal shear stress.

**Figure 6 pharmaceutics-15-00725-f006:**
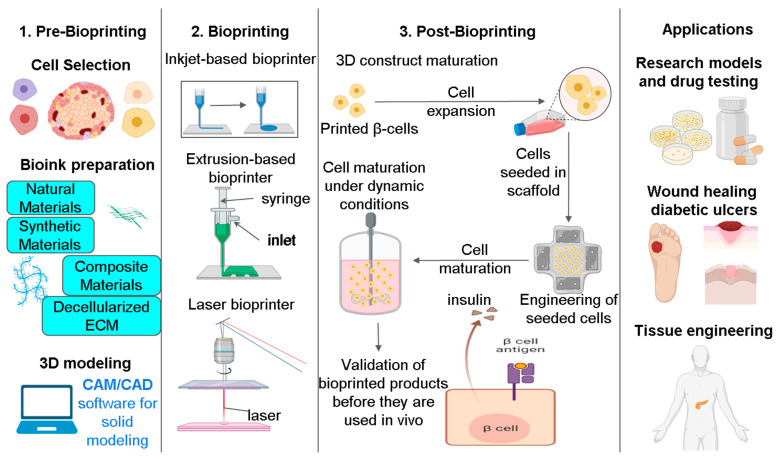
3D bioprinting in diabetic research: 3D bioprinting is used for the fabrication of biocompatible materials, cells, and supporting components to create a functioning living tissue. This process involves: (1) Pre-bioprinting, which consists of the selection of appropriate cells, preparation of bio-ink according to the construct, as well as the setup for computerized assistance to design the model to print; (2) Bioprinting process, where one of the three major methods—inkjet-based printing, extrusion-based printing, and laser bioprinting—are employed to build the construct; (3) Post-bioprinting, where the printed product is left to mature under dynamic conditions, and physiological relevance is kept intact so as to achieve a healthy, biologically functional 3D construct.

**Table 2 pharmaceutics-15-00725-t002:** Characteristic features of commonly used 3D culture models.

3D MODEL	FEATURES	ADVANTAGES	DISADVANTAGES
Organoids	Physiological function can be maintained over a longer period of time (from months to years). High-throughput screening is possible. Helps maintain the overall viability and thereby extend the functional lifetime of cells.	Self-assembling structures. Highly stable and expandable. Patient-specific models can be established. Can be used as a therapeutic tool and in cell therapies [53].	Variability and therefore reduced reproducibility. Complex downstream analysis [54].
Organ on chip (OOC)	Replication of in vivo-like environment and architecture. Organoids can be incorporated into OOCs. Integration of microfluidic immunoassays can seamlessly provide data.	Multiple OOCs can be used to co-culture cells from various organs. Investigation of tissue–tissue interaction is possible. Scaling up is feasible. Low operational cost. Rapid mass transfer. Scope for automation.	Limited high-throughput screening. Non-standard protocols [55]. OOCs are still in the primary stages of development.
Spheroids	Cell aggregates that have self-assembled in an environment that prevents them from attaching to a flat surface [56]. Usually established using ultra-low-attachment plates. The formulation of spheroids is attainable due to the presence of membrane proteins (integrins) [57] and proteins found in the extracellular matrix [58].	Can be formed, propagated, and assayed within the same plates [59]. Cells can be easily co-cultured [60]. Highly functional tool for tissue-based assay, probing cell–cell and cell–matrix interactions [61].	Incidence of fluidic-flow-induced shear stress. Non-uniformity in the size of spheroids produced [59].
Bioreactors	Bioreactors can increase the strength and biocompatibility of synthetic vascular grafts by stimulating seeded cells [62]. To develop an effective bioreactor for a bioprocess, comprehensive research on the biological system is required to understand the cells’ physical and chemical environmental requirements.	Uniform mixing and temperature gradient [63]. High speed, convenient accessibility, flexibility of use, and interoperability of devices [64].	Increased risk of contamination. Some limitations in parameters [65] that can increase the volumes of gradient of oxygen, pH, temperature, and nutrients.
Bioprinting	3D structures are created by dispensing layer after layer of bio-ink and biogel, which when allowed to grow in the right conditions, will result in a functioning tissue mimic with normal metabolic activity [66]. Computer-aided design (CAD) is used to generate the model, which depicts the geometry and size of the study tissue [67].	High speed and availability. High cell viability [68] and cell density [69]. Low cost.	If the structure is too complex the by-product is hard to construct and is inefficient [53]. Lack of precision due to low bio-ink viscocity [70].

## Data Availability

Not applicable.

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
