# Peer review of "New Frontiers in Three-Dimensional Culture Platforms to Improve Diabetes Research"

_pharmaceutics, 2023, doi:10.3390/pharmaceutics15030725_

Round 1

Reviewer 1 Report

Authors did a good review on 3D culture platforms to diabetes treatment considering the 3D models offer a marked advantage in mimicking in vivo tissue architecture. Latest innovations have been discussed and the limitations have also been emphasized. But there are some questions need be promoted before the paper can be accepted.

1.     In 2.4 3D Scaffold systems, page 11, “a 3D cell culture model can either be based on a scaffold or in a scaffold-free system”. Authors discussed a lot of scaffold, but not descript the scaffold-free system. Please provide more information about the scaffold-free system.

2.     In page 16, authors list “3.1 Limitations and improvements need for organoid models”. Where are the other contexts such as “3.2, 3.3….”?

3.     From “3. ORGANOID MODELS, 4. SPHEROIDS, 5. ORGAN ON CHIP, 6. 3D BIOPRINTING, 7. BIOREACTORS”, these items can be classified to different levels. For example, different dimensions (ORGANOID MODELS, SPHEROIDS, ORGAN ON CHIP), prepared method (BIOPRINTING) and culture condition (BIOREACTORS). Please rearrange the items according to their internal logic.  

Author Response

We would like to thank the Reviewers for their fruitful comments, which helped us to improve the manuscript further. We have shown our point-by-point responses to all the comments in the revised manuscript, and our responses are addressed in ‘blue’ fonts.

Comments from Reviewer 1

Authors did a good review on 3D culture platforms to diabetes treatment considering the 3D models offer a marked advantage in mimicking in vivo tissue architecture. Latest innovations have been discussed and the limitations have also been emphasized. But there are some questions need be promoted before the paper can be accepted.

Response: We thank the reviewer for appreciating our manuscript and providing additional suggestions to improve the manuscript further. We have carefully addressed to the comments in the revised manuscript.

Comment 1:   In 2.4 3D Scaffold systems, page 11, “a 3D cell culture model can either be based on a scaffold or in a scaffold-free system”. Authors discussed a lot of scaffold, but not descript the scaffold-free system. Please provide more information about the scaffold-free system.

Response: As per the suggestion of the reviewer, information on scaffold-free system has been included in the revised manuscript on Page 19.

Comment 2:   In page 16, authors list “3.1 Limitations and improvements need for organoid models”. Where are the other contexts such as “3.2, 3.3….”?

Response: The subheading “Limitations and improvements need for organoid models” is now removed, and the content is integrated into the main text of Organoid models in the revised manuscript (Page 16). We thank the reviewer for the suggestion.

Comment 3:   From “3. ORGANOID MODELS, 4. SPHEROIDS, 5. ORGAN ON CHIP, 6. 3D BIOPRINTING, 7. BIOREACTORS”, these items can be classified to different levels. For example, different dimensions (ORGANOID MODELS, SPHEROIDS, ORGAN ON CHIP), prepared method (BIOPRINTING) and culture condition (BIOREACTORS). Please rearrange the items according to their internal logic.  

Response: As suggested by the reviewer, the various 3D models have now been reclassified in the revised manuscript based on scaffold-based and scaffold-free systems.

We thank the reviewer for all the comments and suggestions.

Reviewer 2 Report

This review provides, in my opinion, a fairly exhaustive critical analysis of the advantages and disadvantages of the existing experimental in vivo and vitro models of T1DM and T2MD. There are no fundamental comments. One technical note: in the Abstract replace DM with Diabetes mellitus.

Author Response

This review provides, in my opinion, a fairly exhaustive critical analysis of the advantages and disadvantages of the existing experimental in vivo and vitro models of T1DM and T2MD. There are no fundamental comments. One technical note: in the Abstract replace DM with Diabetes mellitus.

Response: We thank the reviewer for the positive comments and suggestions. As pointed out by the reviewer, DM has been replaced with Diabetes mellitus in the abstract of the revised manuscript.

Reviewer 3 Report

Pharmaceutics

Manuscript by  

Mohanddas et al, “New frontiers of three-dimensional culture platforms to scale-up diabetes research.”

This is a comprehensive review of the current state of the field of pancreatic beta-cell research in which various 3-D culture systems are being developed to improve research capabilities and decrease reliance on animals. That said, the review begins with a very valuable summary of currently available mouse and rat models of diabetes (Table 1), before laying out the general landscape of commonly used 3D culture models (Table 2) which are then explained in the ensuing material.  Separate sections, each with its own color schematic cartoon to illustrate the principles, are devoted to organoids, spheroids, microfluidics (organ-on-chip), 3D bioprinting, and bioreactors (Figs. 2-6).  The work is well-referenced, very detailed, well written, and represents a valuable contribution as a single resource in which many examples from this interesting field are collected and organized in a single place.  

Several corrections and suggestions (all relatively minor) are listed below, with the page (p) and line number (l) indicated:

Edits

p. 1, l. 1  In the title, replace “scale-up” with “improve”

p. 3, l. 1-2 ‘advancements in’… ‘current applications’

p. 3, l. 33 ‘working cultures, wide’

p. 5, l. 10 ‘developed due to the direct cytotoxic action on b-cells’

p. 5, l. 19 ‘breeding the animals’

p. 5, Table 1 - Disadvantage, •occurrence of insulinitis (not insulinities)

p. 6, LETL rat section, right box ‘The numbers of splenic’

p. 8, top 2 right boxes – They indicate references [40] and [42], but not [41]

p. 9, l. 11 ‘in vivo

p. 9, l. 19 ‘sufficient numbers of’

p. 9, bottom panel Features - ‘Helps maintain’

p. 11, l. 10 “enabled the design of scaffolds’

p. 11, l. 17 – need a number for Chan and Leong reference

p. 12, l. 6 ‘stem cells’

p. 12, l. 21 ‘an insulinoma cell line’

p. 13, l. 3 - shouldn’t this be ‘neither completely known nor’ ?

p. 13, 26 – the reference for Lee needs to be indicated as ‘[80]’

p.15, l. 11 – the reference for Petersen needs to be indicated

p. 15, second paragraph – Winner et al. should be indicated as [80]

p. 15, l. 2 from bottom – Kim and Jang reference number is missing

p. 16, bottom line ‘interactions, Figure 3 [89]. The’

p. 17, Figure legend, l. 5 ‘matrix that influences’

p.18, l. 18-19 ‘these … cultures lack’

p. 18, bottom paragraph, l. 6 – Which Lee et al. is indicated, 91 or 97?

p. 19, l. 3 ‘Essaouiba et al. [95]’

p. 19, l. 15 ‘Bauer et al. [96]’

p. 19, l. 27 – The Feng reference is not indicated. Is it reference 76?

p. 21, l. 7 – Essaouira et al. [95]’

p. 21, bottom paragraph –

p. 22. l. 16 ‘analyze molecular’

p. 22, l. 18 ‘short as 50 ms’

p. 24, Figure legend, lines 2, 4, 6 – (1), (2), (3)’

p. 26, l. 21, 24 ‘bioink’

p.28, Figure legend l. 5-6 – ‘(1), (2), (3), (4)’

Author Response

We would like to thank the Reviewers for their fruitful comments, which helped us to improve the manuscript further. We have shown our point-by-point responses to all the comments in the revised manuscript, and our responses are addressed in ‘blue’ fonts.

Comments from Reviewer 3

This is a comprehensive review of the current state of the field of pancreatic beta-cell research in which various 3-D culture systems are being developed to improve research capabilities and decrease reliance on animals. That said, the review begins with a very valuable summary of currently available mouse and rat models of diabetes (Table 1), before laying out the general landscape of commonly used 3D culture models (Table 2) which are then explained in the ensuing material.  Separate sections, each with its own color schematic cartoon to illustrate the principles, are devoted to organoids, spheroids, microfluidics (organ-on-chip), 3D bioprinting, and bioreactors (Figs. 2-6).  The work is well-referenced, very detailed, well written, and represents a valuable contribution as a single resource in which many examples from this interesting field are collected and organized in a single place.  

Several corrections and suggestions (all relatively minor) are listed below, with the page (p) and line number (l) indicated:

Edits

Edit 1:   p. 1, l. 1 In the title, replace “scale-up” with “improve”

Response: As per reviewer’s suggestion, “scale-up” is now replaced with “improve” in the tittle of the revised manuscript.

Edit 2:   p. 3, l. 1-2 ‘advancements in’… ‘current applications’

Response: As suggested, the sentence is now rephrased in the revised manuscript (page 3, l. 2-3).

Edit 3:   p. 3, l. 33 ‘working cultures, wide’

Response: Thanks, we have corrected this in the revised manuscript (page 3, l. 33).

Edit 4:   p. 5, l. 10 ‘developed due to the direct cytotoxic action on b-cells’

Response: The grammatical error is now rectified in the revised manuscript (page 5, l. 9-11).

Edit 5:   p. 5, l. 19 ‘breeding the animals’

Response: The suggested correction is now made in the revised manuscript (page 5, l. 19).

Edit 6:   p. 5, Table 1 - Disadvantage, •occurrence of insulinitis (not insulinities)

Response: The spelling error is now rectified in the revised manuscript (page 5, table 1).

Edit 7:   p. 6, LETL rat section, right box ‘The numbers of splenic’

Response: The suggested grammatical error is now rectified in the revised manuscript (page 6, LETL rat section).

Edit 8:   p. 8, top 2 right boxes – They indicate references [40] and [42], but not [41]

Response: Thanks, we have indicated reference number 41, and corrected the references in the revised manuscript (page 8).

Edit 9:   p. 9, l. 11 ‘in vivo

Response: Thanks, the words in vivo is now italicized (page 9, l. 9) throughout the revised manuscript.

Edit 10:   p. 9, l. 19 ‘sufficient numbers of’

Response: The grammatical mistake is now rectified in the revised manuscript (page 9, l. 17).

Edit 11:   p. 9, bottom panel Features - ‘Helps maintain’

Response: As suggested, the grammatical error is now rectified in the revised manuscript (page 9).

Edit 12:   p. 11, l. 10 “enabled the design of scaffolds’

Response: As pointed out by the reviewer, the grammatical error is now rectified in the revised manuscript (page 11, l. 8-9).

Edit 13:   p. 11, l. 17 – need a number for Chan and Leong reference

Response: As highlighted by the reviewer, the missing citation for Chan and Leong (now reference number ‘71’) is added in the revised manuscript (page 11, l. 14).

Edit 14:   p. 12, l. 6 ‘stem cells’

Response: Thanks, the grammatical mistake is now rectified in the revised manuscript (page 12, l. 5).

Edit 15:   p. 12, l. 21 ‘an insulinoma cell line’

Response: As per the reviewer’s suggestion, the grammatical error is now rectified in the revised manuscript (page 12, l. 20).

Edit 16:   p. 13, l. 3 - shouldn’t this be ‘neither completely known nor’?

Response: As highlighted by the reviewer, the sentence is now rephrased in the revised manuscript (page 13, l. 3-4).

Edit 17:   p. 13, 26 – the reference for Lee needs to be indicated as ‘[80]’

Response: The missing citation for Lee is now added to the end of the sentence in the revised manuscript (page 13, l. 26 and 32).

Edit 18:   p.15, l. 11 – the reference for Petersen needs to be indicated

Response: Thanks, the reference number ‘82’ is now indicated at the end of the sentence (page 15, l. 17).

Edit 19:   p. 15, second paragraph – Winner et al. should be indicated as [80]

Response: As suggested, reference number ‘83’ is now indicated at the end of the sentence for Wimmer et al (page 15, second paragraph).

Edit 20:   p. 15, l. 2 from bottom – Kim and Jang reference number is missing

Response: As pointed out by the reviewer, the missing citation is indicated as reference number ‘26’ in the revised manuscript (page 15, third paragraph).

Edit 21:   p. 16, bottom line ‘interactions, Figure 3 [89]. The’

Response: The correction is made in the revised manuscript (page 20, last line).

Edit 22:   p. 17, Figure legend, l. 5 ‘matrix that influences’

Response: The grammatical error is now corrected in the revised manuscript (Page 21, figure legend 4, l. 5).

Edit 23:   p.18, l. 18-19 ‘these … cultures lack’

Response: As highlighted by the reviewer, the correction is now made in the revised manuscript (page 22, l. 20).

Edit 24:   p. 18, bottom paragraph, l. 6 – Which Lee et al. is indicated, 91 or 97?

Response: The citation is now indicated as number ‘110’ in the revised manuscript. The specific authors are indicted in the revised manuscript (page 22, third paragraph).

Edit 25:   p. 19, l. 3 ‘Essaouiba et al. [95]’

Response: The citation for Essaouiba et al is now added, and indicated as reference number ‘114’ in the revised manuscript (page 23, l. 4).

Edit 26:   p. 19, l. 15 ‘Bauer et al. [96]’

Response: As per the suggestion of the reviewer, the missing citation has been added and indicated as reference ‘115’ in the revised manuscript (page 23, l. 17).

Edit 27:   p. 19, l. 27 – The Feng reference is not indicated. Is it reference 76?

Response: As highlighted by the reviewer, the citation for Feng is now added, and has been indicated as reference number ‘77’ in the revised manuscript (page 23, 7th line from bottom).

Edit 28:   p. 21, l. 7 – Essaouira et al. [95]’

Response: The citation for ‘Essaouiba and Okitsu et al’ is now indicated as reference number ‘88’ in the revised manuscript (page 18, l. 7).

Edit 29:   p. 21, bottom paragraph –

Response: The citations have been indicated as ‘93-95’ in the revised manuscript (page 18, bottom paragraph l. 3).

Edit 30:   p. 22. l. 16 ‘analyze molecular’

Response: Thanks, the spelling error is corrected in the revised manuscript (page 19, l. 11).

Edit 31:   p. 22, l. 18 ‘short as 50 ms’

Response: The mistake is now rectified in the revised manuscript (page 19, l. 13).

Edit 32:   p. 24, Figure legend, lines 2, 4, 6 – (1), (2), (3)’

Response: Thanks, the numbering style in the figure legend is now modified as suggested, in the revised manuscript (page 28, figure legend 6, l. 3, 5, 7).

Edit 33:   p. 26, l. 21, 24 ‘bioink’

Response: As pointed out, the error is now rectified throughout the revised manuscript (page 30, l. 22, 25).

Edit 34:   p.28, Figure legend l. 5-6 – ‘(1), (2), (3), (4)’

Response: Following the suggestion of the reviewer, the numbering style in the figure legend is modified in the revised manuscript (page 25, figure legend 5, l. 5-6).

We thank the reviewer for all the comments and suggestions.